# TAG: Task-based Accumulated Gradients for Lifelong learning

## Abstract

When an agent encounters a continual stream of new tasks in the lifelong learning setting, it leverages the knowledge it gained from the earlier tasks to help learn the new tasks better. In such a scenario, identifying an efficient knowledge representation becomes a challenging problem. Most research works propose to either store a subset of examples from the past tasks in a replay buffer, dedicate a separate set of parameters to each task or penalize excessive updates over parameters by introducing a regularization term. While existing methods employ the general task-agnostic stochastic gradient descent update rule, we propose a task-aware optimizer that adapts the learning rate based on the relatedness among tasks. We utilize the directions taken by the parameters during the updates by additively accumulating the gradients specific to each task. These task-based accumulated gradients act as a knowledge base that is maintained and updated throughout the stream. We empirically show that our proposed adaptive learning rate not only accounts for catastrophic forgetting but also exhibits knowledge transfer. We also show that our method performs better than several state-of-the-art methods in lifelong learning on complex datasets. Moreover, our method can also be combined with the existing methods and achieve substantial improvement in performance.

## 1 Introduction

Lifelong learning (LLL), also known as continual learning, is a setting where an agent continuously learns from data belonging to different tasks (Parisi et al., 2019). Here, the goal is to maximize performance on all the tasks arriving in a stream without replaying the entire datasets from past tasks (Riemer et al., 2018). Approaches proposed in this setting involve investigating the stability-plasticity dilemma (Mermillod et al., 2013) in different ways where stability refers to preventing the forgetting of past knowledge and plasticity refers to accumulating new knowledge by learning new tasks (Mermillod et al., 2013; De Lange et al., 2019).

Unlike human beings, who can efficiently assess the correctness and applicability of the past knowledge (Chen & Liu, 2018), neural networks and other machine learning models often face various issues in this setting. Whenever data from a new task arrives, these models often tend to forget the previously obtained knowledge due to dependency on the input data distribution, limited capacity, diversity among tasks, etc. This leads to a significant drop in performance on the previous tasks - also known as *catastrophic forgetting* (McCloskey & J. Cohen, 1989; Robins, 1993).

Recently there has been an ample amount of research proposed in LLL (De Lange et al., 2019). Several methods, categorized as *Parameter Isolation* methods, either freeze or add a set of parameters as their task knowledge when a new task arrives. Another type of methods, known as *Regularization-based* methods, involve an additional regularization term to tackle the stability-plasticity dilemma. There are approaches based on approximate Bayesian inference, where parameters are sampled from a distribution, that suggest

controlling the updates based on parameter uncertainty (Blundell et al., 2015; Adel et al., 2019; Ahn et al., 2019). But these approaches are computationally expensive and often depend on the choice of prior (Zenke et al., 2017; Nguyen et al., 2018).

Another class of methods, namely *Replay-based* methods, store a subset of examples from each task in a replay buffer. These methods apply gradient-based updates that facilitate a high-level transfer across different tasks through the examples from the past tasks that are simultaneously available while training. As a result, these methods tend to frame LLL as an i.i.d. setting. While replay-based methods are currently state-of-the-art in several LLL tasks, it is important that we explore various ways to tackle the original non-i.i.d. problem (Hadsell et al., 2020). Hence the focus of this paper is to design efficient replay-free methods for LLL.

While adaptive gradient descent based optimizers such as Adam (Kingma & Ba, 2014) have shown superior performance in the classical machine learning setup, many existing works in LLL employ the conventional stochastic gradient descent for parameter update (Lopez-Paz & Ranzato, 2017; Chaudhry et al., 2019; Farajtabar et al., 2020). Adaptive gradient descent based optimizers accumulate gradients to regulate the magnitude and direction of updates but often struggle when the dataset arrives in a non-i.i.d. manner, from different tasks, etc. These optimizers tend to 'over-adapt' on the most recent batch and hence suffer from poor generalization performance (Keskar & Socher, 2017; Chen et al., 2018; Mirzadeh et al., 2020).

While exploiting the adaptive nature of the gradient descent based optimizers, we alleviate catastrophic forgetting by introducing Task-based Accumulated Gradients (*TAG*) that is a wrapper around existing optimizers. The key contributions of our work are as follows:

- We define a task-aware adaptive learning rate for the parameter update step that is also aware of relatedness among tasks in LLL.
- As the knowledge base, we propose to additively accumulate the directions (or gradients) that the network took while learning a specific task instead of storing past examples.
- We empirically show that our method prevents catastrophic forgetting and exhibits knowledge transfer if the tasks are related without introducing more parameters.

Our proposed method, described in Section 3, achieves state-of-the-art results and outperforms several Replay-free methods on complex datasets like **Split-miniImageNet**, **Split-CUB**, etc. For smaller episodic memory, our method also outperforms the *Replay-based* methods as shown in Section 4. Note that we propose a new approach to optimizing based on the past gradients and as such it could potentially be applied along with the existing LLL methods including replay-based methods. We demonstrate the effectiveness of doing the same in the experiments.

## 2 RELATED WORK

Methods proposed in LLL are broadly categorized into three classes: *Regularization-based*, *Parameter Isolation* and *Replay-based* methods (De Lange et al., 2019; Masana et al., 2020). *Regularization-based* methods prevent a drastic change in the network parameters as the new task arrives to mitigate forgetting. These methods further are classified as data-focused (Li & Hoiem, 2017; Triki et al., 2017) and prior-focused methods (Nguyen et al., 2018; Ebrahimi et al., 2019). In particular, Elastic Weight Consolidation (*EWC*) (Kirkpatrick et al., 2017), a prior-focused method, regularizes the loss function to minimize changes in the parameters important for previous tasks. Yet, when the model needs to adapt to a large number of tasks, the interference between task-based knowledge is inevitable with fixed model capacity. *Parameter Isolation* methods (Rusu et al., 2016; Xu & Zhu, 2018; Serra et al., 2018) such as (Aljundi et al., 2017) assign a model copy to every new task that arrives. These methods alleviate catastrophic forgetting in general, but they rely on a strong base network and work on a small number of tasks. Another closely related methods, called Expansion-based methods, handle the LLL problem by expanding the model capacity in order to

adapt to new tasks (Sodhani et al., 2018; Rao et al., 2019). Li et al. (2019) propose to learn task-specific model structures explicitly while retaining model primitives sharing, decoupling from model parameter estimation. *Replay-based* methods maintain an 'episodic memory', containing a few examples from past tasks, that is revisited while learning a new task (Riemer et al., 2018; Jin et al., 2020). For instance, Averaged Gradient Episodic Memory (*A-GEM*) (Chaudhry et al., 2018b), alleviating computational inefficiency of GEM (Lopez-Paz & Ranzato, 2017), uses the episodic memory to project the gradients based on hard constraints defined on the episodic memory and the current mini-batch. Experience Replay (*ER*) (Chaudhry et al., 2019) uses both replay memory and input mini-batches in the optimization step by averaging their gradients to mitigate forgetting.

Task-relatedness (Li & Hoiem, 2017; Jerfel et al., 2019; Shaker et al., 2020) or explicitly learning task representations (Yoon et al., 2017) is also an alternative approach studied in LLL. Efficient Lifelong Learning Algorithm (*ELLA*) (Ruvolo & Eaton, 2013) maintains sparsely shared basis vectors for all the tasks and refines them whenever the model sees a new task. Rao et al. (2019) perform dynamic expansion of the model while learning task-specific representation and task inference within the model. Orthogonal Gradient Descent (*OGD*) (Farajtabar et al., 2020) maintains a space based on a subset of gradients from each task. As a result, *OGD* often faces memory issues during run-time depending upon the size of the model and the subset (Bennani & Sugiyama, 2020). Unlike *OGD*, we accumulate the gradients and hence alleviate the memory requirements by orders for each task.

A recent work (Mirzadeh et al., 2020) argues that tuning the hyper-parameters gives a better result than several state-of-the-art methods including *A-GEM* and *ER*. They introduce *Stable SGD* that involves an adjustment in the hyper-parameters like initial learning rate, learning rate decay, dropout, and batch size. They present this gain in performance on simplistic benchmarks like **Permuted MNIST** (Goodfellow et al., 2013), **Rotated MNIST** and **Split-CIFAR100** (Mirzadeh et al., 2020). Another related work (Gupta et al., 2020) also employs an adaptive learning rate while requiring a small episodic memory. But it is based on a meta-learning setting and hence beyond the scope of our paper.

# 3 METHOD

## 3.1 LIFELONG LEARNING SETUP

In this section, we introduce the notations and the LLL setup used in the paper. We focus on the standard task-incremental learning scenario which is adopted in the numerous state-of-the-art LLL methods. It involves solving new tasks using an artificial neural network with a multi-head output where each head is associated with a unique task and the task identity is known beforehand (Lopez-Paz & Ranzato, 2017; van de Ven & Tolias, 2019). We denote the current task as $t$ and any of the previous tasks by $\tau$. The model receives new data of the form $\{X^{(t)}, D^{(t)}, Y^{(t)}\}$ where $X^{(t)}$ are the input features, $D^{(t)}$ is the task descriptor (that is a natural number in this work) and $Y^{(t)}$ is the target vector specific to the task $t$.

We consider the 'single-pass per task' setting in this work following (Lopez-Paz & Ranzato, 2017; Riemer et al., 2018; Chaudhry et al., 2019). It is more challenging than the multiple pass setting used in numerous research works (Kirkpatrick et al., 2017; Rebuffi et al., 2017). The goal is to learn a classification model $f(X^{(t)}; \theta)$, parameterized by $\theta \in \mathbb{R}^P$ to minimize the loss $L(f(X^{(t)}; \theta), Y^{(t)})$ for the current task $t$ while preventing the loss on the past tasks from increasing. We evaluate the model on a held-out set of examples of all the tasks ($\leq t$) seen in the stream.

## 3.2 TASK-BASED ACCUMULATED GRADIENTS

The specific form of our proposed method depends on the underlying adaptive optimizer. For ease of exposition, we describe it as a modification of RMSProp (Tieleman & Hinton, 2012) here and call it *TAG-*

*RMSProp.* The *TAG* versions of other methods such as the Adagrad (Duchi et al., 2011) and Adam are available in Appendix A.1. A *Naive RMSProp* update, for a given learning rate $\eta$, looks like the following:

$$V_n = \beta V_{n-1} + (1-\beta)g_n^2; \ \ 1 \le n \le N$$
$$\theta_{n+1} = \theta_n - \frac{\eta}{\sqrt{V_n + \epsilon}}g_n \tag{1}$$

where $\theta_n$ is the parameter vector at step $n$ in the epoch, $g_n$ is the gradient of the loss, $N$ is the total number of steps in one epoch, $V_n$ is the moving average of the square of gradients (or the *second moment*), and $\beta$ is the decay rate. We will use *TAG-optimizers* as a generic terminology for the rest of the paper.

We maintain the second moment $V_n^{(t)}$ for each task $t$ in the stream and store it as the knowledge base. When the model shifts from one task to another, the new loss surface may look significantly different. We argue that by using the task-based second moment to regulate the new task updates, we can reduce the interference with the previously optimized parameters in the model. We define the second moment $V_n^{(t)}$ for task $t$ for *TAG-RMSProp* as: $V_n^{(t)} = \beta_2 V_{n-1}^{(t)} + (1-\beta_2)g_n^2$ where $\beta_2$ is constant throughout the stream. We use $\mathbf{V}_n^{(t)}$ to denote a matrix that stores the second moments from all previous tasks, i.e., $\mathbf{V}_n^{(t)} = \{V_N^{(1)}, ..., V_N^{(t-1)}, V_n^{(t)}\}$ of size $(t \times P)$. Hence, the memory required to store these task-specific accumulated gradients increases linearly as the number of tasks in the setting. Note that each $V_N^{(\tau)}$ (where $\tau < t$) vector captures the gradient information when the model receives data from a task $\tau$ and does not change after the task $\tau$ is learned. It helps in regulating the magnitude of the update while learning the current task $t$. To alleviate the catastrophic forgetting problem occurring in the *Naive RMSProp*, we replace $V_n$ (in Eq. 1) to a weighted sum of $\mathbf{V}_n^{(t)}$. We propose a way to regulate the weights corresponding to $\mathbf{V}_n^{(t)}$ for each task in the next section.

### 3.3 ADAPTIVE LEARNING RATE

Next, we describe our proposed learning rate that adapts based on the relatedness among tasks. We discuss how task-based accumulated gradients can help regulate the parameter updates to minimize catastrophic forgetting and transfer knowledge.

We first define a representation for each task to enable computing correlation between different tasks. We take inspiration from a recent work (Guiroy et al., 2019) which is based on a popular meta-learning approach called Model-Agnostic Meta-Learning (MAML) (Finn et al., 2017). Guiroy et al. (2019) suggest that with reference to given parameters $\theta^s$ (where $s$ denotes shared parameters), the similarity between the adaptation trajectories (and also meta-test gradients) among the tasks can act as an indicator of good generalization. This similarity is defined by computing the inner dot product between adaptation trajectories. In the experiments, Guiroy et al. (2019) show an improvement in the overall target accuracy by adding a regularization term in the outer loop update to enhance the similarity.

In case of LLL, instead of a fixed point of reference $\theta^s$, the parameters continue to update as the model adapts to a new task. Analogous to the adaptation trajectories, we essentially want to capture those task-specific *gradient directions* in the LLL setting. Momentum serves as a less noisy estimate for the overall gradient direction and hence approximating the adaptation trajectories. The role of momentum has been crucial in the optimization literature for gradient descent updates (Ruder, 2016; Li et al., 2017). Therefore, we introduce the task-based *first moment* $M_n^{(t)}$ in order to approximate the adaptation trajectories of each task $t$. It is essentially the momentum maintained while learning each task $t$ and would act as the task representation for computing the correlation.

The $M_n^{(t)}$ is defined as: $M_n^{(t)} = \beta_1 M_{n-1}^{(t)} + (1-\beta_1)g_n$ where $\beta_1$ is the constant decay rate. Intuitively, if the current task $t$ is correlated with a previous task $\tau$, the learning rate in the parameter update step should be higher to encourage the transfer of knowledge between task $t$ and $\tau$. In other words, it should allow

knowledge transfer. Whereas if the current task $t$ is uncorrelated or negatively correlated to a previous task $\tau$, the new updates over parameters may cause catastrophic forgetting because these updates for task $t$ may point in the opposite direction of the previous task $\tau$'s updates. In such a case, the learning rate should adapt to lessen the effects of the new updates. We introduce a scalar quantity $\alpha_n(t, \tau)$ to capture the correlation that is computed using $M_n^{(t)}$ and $M_N^{(\tau)}$:

$$\alpha_n(t, \tau) = exp(-b \; \frac{M_n^{(t)^T} M_N^{(\tau)}}{|M_n^{(t)}||M_N^{(\tau)}|})$$

(2)

where $|.|$ is the Euclidean norm. Existing adaptive optimizers, such as Adam, tend to overfit on the most recent dataset from a task, which results in catastrophic forgetting in LLL. By using the exponential term, the resulting $\alpha_n(t, \tau)$ will attain a higher value for uncorrelated tasks and will minimize the new updates (hence prevent forgetting). Here, $b$ is a hyperparameter that tunes the magnitude of $\alpha_n(t, \tau)$. The higher the $b$ is, the greater is the focus on preventing catastrophic forgetting. Its value can vary for different datasets. For the current task $t$ at step $n$ (with $\theta_1^{(t)} = \theta_{N+1}^{(t-1)}$), we define the *TAG-RMSProp* update as:

$$\theta_{n+1}^{(t)} = \theta_n^{(t)} - \frac{\eta}{\sqrt{\alpha_n(t, t) \, V_n^{(t)} + \sum\limits_{\tau=1}^{t-1} \alpha_n(t, \tau) \, V_N^{(\tau)} + \epsilon}} g_n$$

(3)

Hence, the role of each $\alpha_n(t, \tau)$ is to regulate the influence of corresponding task-based accumulated gradient $V_N^{(\tau)}$ of the previous task $\tau$. Since we propose a new way of looking at the gradients, our update rule (Eq. 3) can be applied with any kind of task-incremental learning setup. In this way, the overall structure of the algorithm for this setup remains the same.

## 4  EXPERIMENTS

We describe the experiments performed to evaluate our proposed method.[1] In the first experiment, we show the gain in performance by introducing *TAG* update instead of naive optimizers update. We analyse how our proposed learning rate adapts and achieves a higher accuracy over the tasks in the stream. Next, we compare our proposed replay-free method with other state-of-the-art baselines and also show that *TAG* update (in Eq. 3) can be used along with other state-of-the-art methods to improve their results.

The experiments are performed on four benchmark datasets: **Split-CIFAR100**, **Split-miniImageNet**, **Split-CUB** and **5-dataset**. **Split-CIFAR100** and **Split-miniImageNet** splits the **CIFAR-100** (Krizhevsky et al., 2009; Mirzadeh et al., 2020) and **Mini-imagenet** (Vinyals et al., 2016; Chaudhry et al., 2019) datasets into 20 disjoint 5-way classification tasks. **Split-CUB** splits the **CUB** (Wah et al., 2011) dataset into 20 disjoint tasks with 10 classes per task. **5-dataset** is a sequence of five different datasets as five 10-way classification tasks. These datasets are: **CIFAR-10** (Krizhevsky et al., 2009), **MNIST** (LeCun, 1998), **SVHN** (Netzer et al., 2011), **notMNIST** (Bulatov, 2011) and **Fashion-MNIST** (Xiao et al., 2017). More details about the datasets are given in Appendix A.2. For experiments with **Split-CIFAR100** and **Split-miniImageNet**, we use a reduced ResNet18 architecture following (Lopez-Paz & Ranzato, 2017; Chaudhry et al., 2019). We use the same reduced ResNet18 architecture for **5-dataset**. For **Split-CUB**, we use a ResNet18 model which is pretrained on Imagenet dataset (Deng et al., 2009) as used in (Chaudhry et al., 2019).

We report the following metrics by evaluating the model on the held-out test set: **(i) Accuracy** (Lopez-Paz & Ranzato, 2017) i.e., average test accuracy when the model has been trained sequentially up to the latest task,

---

[1]Code for the experiments is submitted as supplementary material and will be released publicly upon acceptance.

**(ii) Forgetting** (Chaudhry et al., 2018a) i.e., decrease in performance of each task from their peak accuracy to their accuracy after training on the latest task and **(iii) Learning Accuracy (LA)** (Riemer et al., 2018) i.e., average accuracy for each task immediately after it is learned. The overall goal is to maximise the average test **Accuracy**. Further, a LLL algorithm should also achieve high **LA** while maintaining a low value of **Forgetting** because it should learn the new task better without compromising its performance on the previous tasks (see Appendix A.2).

We report the above metrics on the best hyper-parameter combination obtained from a grid-search. The overall implementation of the above setting is based on the code provided by (Mirzadeh et al., 2020). The details of the grid-search and other implementation details corresponding to all experiments described in our paper are given in Appendix A.2.1. For all the experiments described in this section, we train the model for a single epoch per task. Results for multiple epochs per task are given in Appendix A.3.4. All the performance results reported are averaged over five runs.

### 4.1 NAIVE OPTIMIZERS

We validate the improvement by our proposed setting over the gradient descent based methods and demonstrate the impact of using correlation among tasks in the *TAG-optimizers*. Firstly, we train the model on a stream of tasks using *Naive SGD* update without applying any specific LLL method. Similarly, we replace the SGD update with *Naive Adagrad*, *Naive RMSProp*, *Naive Adam* and their respective *TAG-optimizers* to compare their performances. We show the resulting **Accuracy (%)** (in Fig. 1a) and **Forgetting** (in Fig. 1b) when the model is trained in with the above-mentioned optimizers.

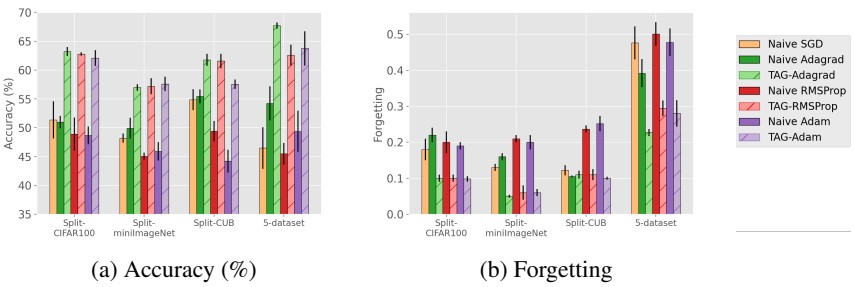

(a) Accuracy (%)  (b) Forgetting

Figure 1: Final average test **Accuracy (%)** (higher is better) and **Forgetting** (lower is better) obtained after the stream is finished for all four datasets. The vertical bars with hatches are the performance by *TAG-optimizers* while others are *Naive* optimizers. *TAG-optimizers* outperforms the naive optimizers in all datasets in terms of accuracy and also results in a lower forgetting.

It is clear that *TAG-optimizers* outperform their naive counterparts as well as *Naive SGD* for all four datasets by a significant amount. There is a notable decrease in **Forgetting** by *TAG-optimizers* (in Fig. 1b) in general that eventually reflects on the gain in final test **Accuracy** as seen in Fig. 1a. In **Split-CUB**, *TAG-Adam* (57%) shows a remarkable improvement in accuracy when compared to *Naive Adam* (45%) such that it even surpasses *Naive SGD* (55%). Interestingly, *TAG-Adam* results in slightly lower accuracy as compared to *TAG-Adagrad* except in **Split-miniImageNet**. Moreover, *Naive Adagrad* results in a better performance than *Naive RMSProp* and *Naive Adam* for all the datasets. This observation aligns with the results by Hsu et al. (2018). *Naive SGD* performs almost equivalent to *Naive Adagrad* except in **5-dataset** where it is outperformed.

Next, we analyse $\alpha(t, \tau)$ which is the average of $\alpha_n(t, \tau)$ across all steps $n$ for all $t$ and $\tau$ when stream is finished i.e., $\alpha(t, \tau) = \frac{1}{N} \sum_{n=1}^{N} \alpha_n(t, \tau)$. We show how $\alpha(t, \tau)$ values play role in the gain in *TAG-*

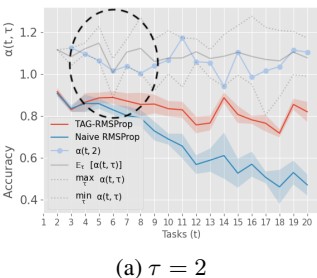 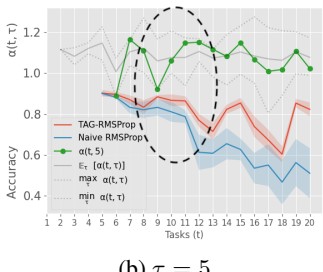 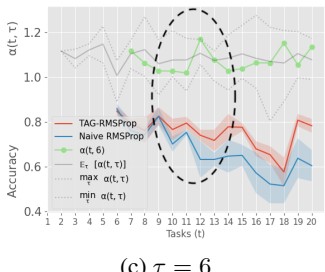

(a) $\tau = 2$         (b) $\tau = 5$         (c) $\tau = 6$

Figure 2: Evolution of (a) $\alpha(t, 2)$ and test accuracy $a_{t,2}$ (left), (b) $\alpha(t, 5)$ and test accuracy $a_{t,5}$ (middle) and (c) $\alpha(t, 6)$ and test accuracy $a_{t,6}$ (right) along the stream of 20 tasks in the **Split-CUB** dataset. The grey-coloured lines are $\max_{\tau'} \alpha_n(t, \tau')$ (top, dashed line), $\mathbb{E}_{\tau'}[\alpha(t, \tau')]$ (middle, solid line) and $\min_{\tau'} \alpha(t, \tau')$ (bottom, dashed line) that indicate the range of $\alpha(t, \tau')$. Elliptical regions (black dashed) highlight subtle gain in the accuracy by *TAG-RMSProp* that are maintained throughout the stream. Observing corresponding $\alpha(t, \tau)$ in those regions validates our hypothesis discussed from Section 3.3.

*optimizers* accuracies in case of **Split-CUB** dataset. Each plot in Fig. 2 corresponds to test accuracies $a_{t,\tau}$ (with shaded areas indicating their standard deviations) for *Naive RMSProp* (blue) and *TAG-RMSProp* (red) for a particular $\tau$ for all tasks $t$ in the stream (x-axis). Along with that, the grey-coloured curves are $\max_{\tau'} \alpha_n(t, \tau')$ (top, dashed line), $\mathbb{E}_{\tau'}[\alpha(t, \tau')]$ (middle, solid line) and $\min_{\tau'} \alpha(t, \tau')$ (bottom, dashed line) respectively. These are shown along with the corresponding $\alpha(t, \tau)$ to indicate the rank of $\alpha(t, \tau)$ in the set $\{\alpha(t, \tau'); \ \tau' \in [1, t]\}$ computed the model encounters a task $t$ in the stream.

The accuracies of *TAG-RMSProp* and *Naive RMSProp* appear to correlate for most of the stream. We mark the regions (black dashed ellipse) of subtle improvements in the accuracy by *TAG-RMSProp* that is later maintained throughout the stream. While observing $\alpha(t, \tau)$ in those regions particularly, we note that the rank of $\alpha(t, \tau)$ affects the accuracy in *TAG-RMSProp* as following: **(i)** Lower (or decrease in) rank of $\alpha(t, \tau)$ means that there exists some correlation between the tasks $t$ and $\tau$. So, the model should take advantage of the current task updates and seek (or even amplify) backward transfer. Such observations can be made for the following $(t, \tau)$: $(4, 2), (5, 2), (14, 2), (9, 5), (14, 6), (19, 6)$ etc. Our method also prevents drastic forgetting as well in few cases. For example: $(7, 2), (12, 2), (10, 6)$. **(ii)** Higher (or increase in) rank of $\alpha(t, \tau)$ results in prevention of forgetting as observed in Fig. 2. Such $(t, \tau)$ pairs are $(11, 2), (7, 5), (12, 5), (12, 6)$, etc. It also results in backward transfer as observed in $(11, 5)$ and $(15, 5)$. We report the same analysis for the other three datasets and show the results in Appendix A.3.3.

## 4.2 COMPARED WITH OTHER BASELINES

In the next experiment, we show that the *TAG-RMSProp* results in a strong performance as compared to other LLL algorithms. In Table 1, we report the performance of *TAG-RMSProp* and the following state-of-the-art baselines: *EWC* (Kirkpatrick et al., 2017), *A-GEM* (Chaudhry et al., 2018b), *ER* (Aljundi et al., 2019) with reservoir sampling and *Stable SGD* (Mirzadeh et al., 2020). Apart from these baselines, we report the performance of *Naive SGD* and *Naive RMSProp* from the previous section. We also report results on multi-task learning (MTL) settings on all four datasets where the dataset from all the tasks is always available throughout the stream. Hence, the resulting accuracies of the MTL setting serve as the upper bounds for the test accuracies in LLL.

Following Mirzadeh et al. (2020), the size of the episodic memory for both *A-GEM* and *ER* is set to store 1 example per class. Since we want to evaluate *TAG* with all other baselines on the original non-i.i.d. problem, we keep the episodic memory size in the replay-based methods small for the comparison. We still report

Table 1: Comparing performance in terms of final average test **Accuracy (%)** (higher is better), **Forgetting** (lower is better) and Learning Accuracy (**LA (%)**) (higher is better) of the existing baselines with *TAG-RMSProp* on all four datasets. All metrics are averaged across 5 runs. Overall, *TAG-RMSProp* outperforms all other methods in terms of **Accuracy**. *MTL* assumes that the whole dataset from all tasks is always available during training, hence it is a different setting and its accuracy acts as an upper bound.

| Methods | Split-CIFAR100 | | | Split-miniImageNet | | |
|---|---|---|---|---|---|---|
| | Accuracy (%) | Forgetting | LA (%) | Accuracy (%) | Forgetting | LA (%) |
| *Naive SGD* | 51.36 (±3.21) | 0.18 (±0.03) | 68.46 (±1.93) | 48.19 (±0.79) | 0.13 (±0.01) | 60.6 (±0.95) |
| *Naive RMSProp* | 48.91 (±2.88) | 0.2 (±0.03) | 67.28 (±0.43) | 45.06 (±0.6) | 0.21 (±0.01) | 64.39 (±1.02) |
| *EWC* | 49.06 (±3.44) | 0.19 (±0.04) | 66.82 (±1.41) | 47.87 (±2.08) | 0.15 (±0.02) | 61.66 (±1.06) |
| *A-GEM* | 54.25 (±2.0) | 0.16 (±0.03) | 68.98 (±1.19) | 50.32 (±1.29) | 0.11 (±0.02) | 61.02 (±0.64) |
| *ER* | 59.14 (±1.77) | 0.12 (±0.02) | 70.36 (±1.23) | 54.67 (±0.71) | 0.1 (±0.01) | 64.06 (±0.41) |
| *Stable SGD* | 57.04 (±1.07) | 0.09 (±0.0) | 64.62 (±0.91) | 51.81 (±1.66) | 0.09 (±0.01) | 59.99 (±0.94) |
| **TAG-RMSProp (Ours)** | **62.79** (±0.29) | 0.1 (±0.01) | 72.06 (±1.01) | **57.2** (±1.37) | 0.06 (±0.02) | 62.73 (±0.61) |
| *MTL** | 67.7 (±0.58) | - | - | 66.14 (±1.0) | - | - |

| Methods | Split-CUB | | | 5-dataset | | |
|---|---|---|---|---|---|---|
| | Accuracy (%) | Forgetting | LA (%) | Accuracy (%) | Forgetting | LA (%) |
| *Naive SGD* | 54.88 (±1.83) | 0.12 (±0.01) | 65.97 (±0.59) | 46.48 (±3.62) | 0.48 (±0.05) | 84.55 (±1.06) |
| *Naive RMSProp* | 49.4 (±1.77) | 0.24 (±0.01) | 71.76 (±0.94) | 45.49 (±1.89) | 0.5 (±0.03) | 85.58 (±1.21) |
| *EWC* | 55.66 (±0.97) | 0.12 (±0.01) | 66.36 (±0.71) | 48.58 (±1.47) | 0.4 (±0.03) | 79.56 (±3.18) |
| *A-GEM* | 56.91 (±1.37) | 0.1 (±0.01) | 65.6 (±0.73) | 55.9 (±2.58) | 0.34 (±0.04) | 82.61 (±2.13) |
| *ER* | 59.25 (±0.82) | 0.1 (±0.01) | 66.17 (±0.42) | 61.58 (±2.65) | 0.28 (±0.04) | 84.3 (±1.08) |
| *Stable SGD* | 53.76 (±2.14) | 0.11 (±0.01) | 62.15 (±1.12) | 46.51 (±2.75) | 0.46 (±0.03) | 83.3 (±1.44) |
| **TAG-RMSProp (Ours)** | **61.58** (±1.24) | 0.11 (±0.01) | 71.56 (±0.74) | **62.59** (±1.82) | 0.29 (±0.02) | 86.08 (±0.55) |
| *MTL** | 71.65 (±0.76) | - | - | 70.0 (±4.44) | - | - |

*A-GEM* and *ER* results with bigger memory sizes in Appendix A.3.5. The size of the mini-batch sampled from the episodic memory is set equal to the batch-size to avoid data imbalance while training. Although we utilize a similar amount of memory as (Rusu et al., 2016), an expansion-based method, we do not make any changes to the size of the model and thus the number of parameters remains the same during the test time. Hence, we do not compare our approach with the expansion-based method in our experiments.

From the results reported in Table 1, we observe that *TAG-RMSProp* achieves the best performance in terms of test **Accuracy** as compared to other baselines for all datasets. The overall improvement is decent in **Split-CIFAR100**, **Split-miniImageNet** and **Split-CUB** which are 3.65%, 2.5% and 2.3% with regard to the next-best baseline. On the other hand, the improvement by *TAG-RMSProp* is relatively minor in **5-dataset** i.e., 1% as compared to *ER* with the similar amount of **Forgetting** (0.29 and 0.28) occurring in the stream. In terms of **LA**, *TAG-RMSProp* achieves almost similar performance as *Naive RMSProp* in **Split-CUB** and **5-dataset**. We also note that the **LA** of *TAG-RMSProp* is higher in **Split-CIFAR100**, **Split-CUB** and **5-dataset** than *ER* and *A-GEM*. The higher **LA** with similar **Forgetting** as compared to other baselines shows that while *TAG* exploits the adaptive nature of existing optimizers, it also ensures minimal forgetting of the gained knowledge. The existing optimizers tend to aggressively fit the model on the most recent task at an immense cost of forgetting the earlier tasks. Hence, even if a similar (or lower) **Forgetting** occurs in *TAG*, the higher test **Accuracy** (with high **LA**) shows that *TAG* is capable of retaining the gained knowledge from each task. Although **LA** is lower in **Split-miniImageNet**, *TAG-RMSProp* manages to prevent catastrophic forgetting better than these methods and hence results in a higher test **Accuracy**.

## 4.3 COMBINED WITH OTHER BASELINES

Lastly, we show that the existing baselines can also benefit from our proposed method *TAG-RMSProp*. We replace the conventional SGD update from *EWC*, *A-GEM* and *ER*, and apply RMSProp update (Eq. 1) and

*TAG-RMSProp* update (Eq. 3) respectively. We use the same task-incremental learning setup as used in the previous sections in terms of architecture and hyper-parameters. We compare the resulting accuracies of the baselines with their *RMSProp* and *TAGed* versions in Fig. 3.

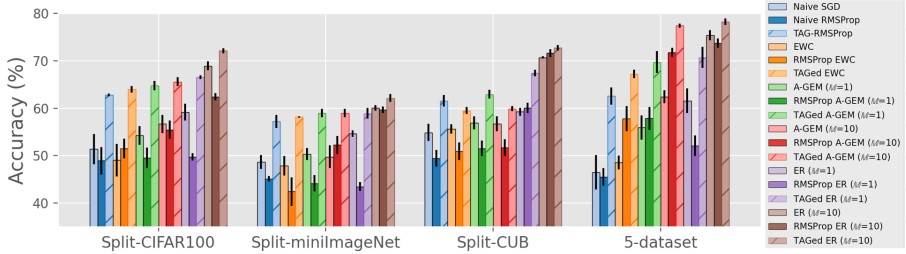

Figure 3: Comparing performance for different existing methods with their *RMSProp* and *TAGed* versions on all four datasets in terms of final average test **Accuracy (%)** along with *A-GEM* and *ER* for different samples per class ($\mathbb{M}$) in the episodic memory. The vertical bars with hatches are the performance by *TAGed* versions of the baselines. All results are averaged across 5 runs. All *TAGed* versions results in a similar gain in the accuracy over baselines with both SGD and RMSProp update.

For a given dataset, we see that gain in the final accuracy in the *TAGed* versions is similar for the baselines described in Section 4.2. That is, *TAG* improves these baselines with SGD update on **Split-CIFAR100**, **Split-miniImageNet**, **Split-CUB** and **5-dataset** by at least 8%, 4%, 4% and 9% respectively. On the other hand, *TAG* improves the baselines with RMSProp update on the datasets by at least 12%, 12%, 7% and 9% respectively. The improvement is also significant in *A-GEM* with bigger episodic memory (i.e., 10 samples per class or $\mathbb{M} = 10$) but we observe relatively smaller improvement (2%) by *TAGed ER* ($\mathbb{M} = 10$) as compared to *ER* ($\mathbb{M} = 10$). These results show that apart from outperforming the baselines independently (with smaller episodic memory in replay-based methods), *TAG* can also be used as an update rule in the existing research works for improving their performances. While *A-GEM* and *ER* are strong baselines for LLL, we would like to highlight that these replay-based methods are not applicable in settings where storing examples is not an option due to privacy concerns. *TAG-RMSProp* would be a more appropriate solution in such settings.

## 5 CONCLUSION

We propose a new task-aware optimizer for the LLL setting that adapts the learning rate based on the relatedness among tasks. We introduce the task-based accumulated gradients that act as the representation for individual tasks for the same. We conduct experiments on complex datasets to compare *TAG-RMSProp* with several state-of-the-art methods. Results show that *TAG-RMSProp* outperforms the existing methods in terms of final accuracy with a commendable margin without storing past examples or using dynamic architectures. We also show that it results in a significant gain in performance when combined with other baselines. To the best of our knowledge, ours is the first work in the LLL literature showing that we can use an adaptive gradient method for LLL and prevent forgetting better than *Naive SGD*. For future work, as the memory required to store the task-specific accumulated gradients increases linearly with the tasks, reducing memory complexity without compromising the performance can be an interesting direction. This can be achieved by **(i)** computing correlation using a smaller quantity than the task-based first moments, and **(ii)** clustering the similar tasks together to reduce the number of task-based second moments (in settings with a soft margin between the tasks). Another possible direction from here can be shifting to a class-incremental scenario where the task identity is not known beforehand and is required to be inferred along the stream.

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

## A APPENDIX

In this document, we provide the details and results excluded from the main paper. In A.1, we describe the *TAG* versions of Adagrad and Adam. The implementation details are described in Section A.2. We also report results obtained by performing additional experiments in Section A.3.

### A.1 TAG-OPTIMIZERS

Similar to RMSProp (in Section 3), with the task-based first moments $M_n^{(t)}$, let $W_n^{(t)} = \{\alpha_n(t, \tau); \tau \in [1, t]\}$ (from Eq. 2) and,

$$
W_n^{(t)} \mathbf{V}_n^{(t)} = \begin{cases} V_n^{(1)}; & t = 1 \\ \alpha_n(t, t) V_n^{(t)} + \sum_{\tau=1}^{t-1} \alpha_n(t, \tau) V_N^{(\tau)}; & t > 1 \end{cases} \tag{4}
$$

We define the *TAG* versions of Adagrad and Adam as following:

- *TAG-Adagrad*:

$$V_n^{(t)} = V_{n-1}^{(t)} + g_n^2$$
$$\theta_{n+1}^{(t)} = \theta_n^{(t)} - \frac{\eta}{\sqrt{W_n^{(t)}\mathbf{V}_n + \epsilon}}g_n \tag{5}$$

- *TAG-Adam*:

$$V_n^{(t)} = \beta_2 V_{n-1}^{(t)} + (1 - \beta_2)g_n^2$$
$$\theta_{n+1}^{(t)} = \theta_n^{(t)} - \frac{\eta\sqrt{1 - \beta_2^n}}{(1 - \beta_1^n)\sqrt{W_n^{(t)}\mathbf{V}_n + \epsilon}}M_n^{(t)} \tag{6}$$

Both *TAG-Adagrad* (Eq. 5) and *TAG-Adam* (Eq. 6) result in a significant gain in Accuracy and prevent Forgetting as observed in Fig. 1a and Fig. 1b respectively in Section 4.1.

## A.2 IMPLEMENTATION DETAILS

The summary of the datasets used in the experiments is shown in Table 2 and Table 3.

Table 2: Dataset Statistics

|  | Input size | Training samples per task | Test samples per task |
|---|---|---|---|
| **Split-CIFAR100** | $3 \times 32 \times 32$ | 2500 | 500 |
| **Split-miniImageNet** | $3 \times 84 \times 84$ | 2400 | 600 |
| **Split-CUB** | $3 \times 224 \times 224$ | 300 | 290 |

In **5-dataset**, we convert all the monochromatic images to RGB format depending on the task dataset. All images are then resized to $3 \times 32 \times 32$. The overall training and test data statistics of **5-dataset** are described in Table 3.

Table 3: **5-dataset** statistics.

|  | Training samples | Test samples |
|---|---|---|
| **CIFAR-10** | 50000 | 10000 |
| **MNIST** | 60000 | 10000 |
| **SVHN** | 73257 | 26032 |
| **notMNIST** | 16853 | 1873 |
| **Fashion-MNIST** | 60000 | 10000 |

Details about the metrics used for evaluating the model:

- **Accuracy** (Lopez-Paz & Ranzato, 2017): If $a_{t,\tau}$ is the accuracy on the test set of task $\tau$ when the current task is $t$, it is defined as, $A_t = \frac{1}{t}\sum_{\tau=1}^{t} a_{t,\tau}$.
- **Forgetting** (Chaudhry et al., 2018a): It is the average forgetting that occurs after the model is trained on all tasks. If the latest task is $t$ and is defined as, $F_t = \frac{1}{t-1}\sum_{\tau=1}^{t-1} \max_{t' \in \{1,...,t-1\}}(a_{t',\tau} - a_{t,\tau})$.

- **Learning Accuracy (LA)** (Riemer et al., 2018): It is the measure of learning capability when the model sees a new task. For the current task $t$, it is defined as, $L_t = \frac{1}{t} \sum_{\tau=1}^{t} a_{\tau,\tau}$.

We implement the following baselines to compare with our proposed method:

- *EWC*: Our implementation of *EWC* is based on the original paper (Kirkpatrick et al., 2017).
- *A-GEM*: We implemented *A-GEM* based on the the official implementation provided by (Chaudhry et al., 2018b).
- *ER* (Chaudhry et al., 2019): Our implementation is based on the one provided by (Aljundi et al., 2019) with reservoir sampling except that the sampled batch does not contain examples from the current task.
- *Stable SGD* (Mirzadeh et al., 2020): We obtain the best hyper-parameter set by performing grid-search over different combinations of the learning rate, learning rate decay, and dropout (see Appendix A.2.1).

*OGD* Farajtabar et al. (2020) requires storing N (=200 in their experiments) number of gradients per task and it is evaluated only on variants of the MNIST dataset by training a small feed-forward network. On the other hand, *TAG* additively accumulates the gradients and hence requires memory equal to two copies of the model as the knowledge base. This enabled us to train a reduced ResNet18 on complex datasets. Due to greater memory requirements, *OGD* faced memory errors in our setting. We would also like to highlight that *OGD* use *Naive-SGD* and our contribution being an adaptive learning rate based method is complementary to this approach.

We provide our code as supplementary material that contains the scripts for reproducing the results from all experiments described in this paper. In the CODE folder, we include README.MD file that contains the overall code structure, procedure for installing the required packages, links to download the datasets and steps to execute the scripts. All experiments were executed on an NVIDIA GTX 1080Ti machine with 11 GB GPU memory.

### A.2.1 HYPER-PARAMETER DETAILS

In this section, we report the grid search details for finding the best set of hyper-parameters for all datasets and baselines. We train the model with $90\%$ of the training set and choose the best hyper-parameters based on the highest accuracy on the validation set which consists of remaining $10\%$ for the training set. For existing baselines, we perform the grid search either suggested by the original papers or by Farajtabar et al. (2020). For all *TAG-optimizers*, $\beta_1$ is set to 0.9. For *TAG-RMSProp* and *TAG-Adagrad*, $\beta_2$ is set to 0.99 and for *TAG-Adam* it is 0.999. In all the experiments, the mini-batch size is fixed to 10 for **Split-CIFAR100**, **Split-miniImageNet**, **Split-CUB** similar to (Chaudhry et al., 2019; Mirzadeh et al., 2020). We set mini-batch size to 64 for **5-dataset** following (Serra et al., 2018). This is because we wanted to highlight the role of learning rate and to show how *TAG-RMSProp* improves the performance while the other hyper-parameters (including batch-size) were fixed.

- *Naive SGD*
  - Learning rate: [0.1 (**Split-CIFAR100**, **5-dataset**), 0.05 (**Split-miniImageNet**), 0.01(**Split-CUB**), 0.001]
- *Naive Adagrad*
  - Learning rate: [0.01, 0.005 (**Split-CIFAR100**, **Split-miniImageNet**, **5-dataset**), 0.001, 0.0005 (**Split-CUB**), 0.0001]

- *Naive RMSProp*
  - Learning rate: [0.01, 0.005 (**Split-CIFAR100**), 0.001 (**Split-miniImageNet**, **5-dataset**), 0.0005, 0.0001 (**Split-CUB**), 0.00005, 0.00001]
- *Naive Adam*
  - Learning rate: [0.01, 0.005 (**Split-CIFAR100**), 0.001 (**Split-miniImageNet**, **5-dataset**), 0.0005, 0.0001 (**Split-CUB**)]
- *TAG-Adagrad*
  - Learning rate: [0.005 (**Split-CIFAR100**, **5-dataset**), 0.001 (**Split-miniImageNet**), 0.0005 (**Split-CUB**), 0.00025, 0.0001]
  - $b$: [1, 3, 5 (**Split-CIFAR100**, **Split-miniImageNet**, **Split-CUB**), 7 (**5-dataset**)]
- *TAG-RMSProp*
  - Learning rate: [0.005, 0.001, 0.0005 (**5-dataset**), 0.00025 (**Split-CIFAR100**), 0.0001 (**Split-miniImageNet**), 0.00005, 0.000025 (**Split-CUB**), 0.00001]
  - $b$: [1, 3, 5 (**Split-CIFAR100**, **Split-miniImageNet**, **Split-CUB**), 7 (**5-dataset**)]
- *TAG-Adam*
  - Learning rate: [0.005, 0.001 (**5-dataset**), 0.0005 (**Split-CIFAR100**), 0.00025 (**Split-miniImageNet**), 0.0001 (**Split-CUB**)]
  - $b$: [1, 3, 5 (**Split-CIFAR100**, **Split-miniImageNet**, **Split-CUB**), 7 (**5-dataset**)]
- *EWC*
  - Learning rate: [0.1 (**Split-CIFAR100**, **5-dataset**), 0.05 (**Split-miniImageNet**), 0.01(**Split-CUB**), 0.001]
  - $\lambda$ (regularization): [1 (**Split-CIFAR100**, **Split-miniImageNet**, **Split-CUB**), 10, 100 (**5-dataset**)]
- *A-GEM*
  - Learning rate: [0.1 (**Split-CIFAR100**, **Split-miniImageNet**, **5-dataset**), 0.05, 0.01(**Split-CUB**), 0.001]
- *ER*
  - Learning rate: [0.1 (**Split-CIFAR100**, **5-dataset**), 0.05 (**Split-miniImageNet**), 0.01(**Split-CUB**), 0.001]
- *Stable SGD*
  - Initial learning rate: [0.1 (**Split-CIFAR100**, **Split-miniImageNet**, **5-dataset**), 0.05 (**Split-CUB**), 0.01]
  - Learning rate decay: [0.9 (**Split-CIFAR100**, **Split-miniImageNet**, **Split-CUB**), 0.8, 0.7 (**5-dataset**)]
  - Dropout: [0.0 (**Split-miniImageNet**, **Split-CUB**, **5-dataset**), 0.1 (**Split-CIFAR100**), 0.25, 0.5]

In case of *TAG-RMSProp*, we empirically found that the best performance of the all three benchmarks with 20 tasks occurred when hyper-parameter $b = 5$ and for **5-dataset**, $b = 7$. We also found that a lower value of Learning rate in *TAG-RMSProp* results in a better performance. These empirical observations can reduce the search space for hyperparameter setup by a huge amount when applying *TAG-RMSProp* on a LLL setup.

For the experiments in Section 4.3 that require a hybrid version of these methods, we use the same hyper-parameters from above except for *TAGed ER* in **Split-CIFAR100** (Learning rate = 0.0005) and **Split-CUB** (Learning rate = 0.0001). We choose the learning rates of *TAG-RMSProp* and *Naive-RMSProp* over *EWC*, *A-GEM* and *ER*.

## A.3 ADDITIONAL EXPERIMENTS

In this section, we describe the additional experiments and analysis done in this work.

### A.3.1 BACKWARD TRANSFER METRIC

While we show the occurrence of knowledge transfer in Fig. 2, we can quantify the Backward Transfer (**BWT**) (Chaudhry et al., 2019) by computing the difference between the final **Accuracy** and **LA**. i.e., $\textbf{BWT} = \frac{1}{t-1} \sum_{\tau=1}^{t-1} a_{t,\tau} - a_{\tau,\tau}$. We report the **BWT** results for all datasets and baselines in Table 4.

While *TAG-RMSProp* outperforms the other baselines in terms of **BWT** for **Split-miniImageNet**, it is overall the second-best method for **Split-CIFAR100** and **5-dataset**. In case of **Split-CUB**, even if **TAG-RMSProp** achieves the highest **Accuracy**, it results in a lower **BWT** because of a significantly higher **LA** as compared to the other baselines (see Table 1).

Table 4: Comparing performance in terms of final average test **Accuracy (%)** (higher is better) and **BWT** (higher is better) with the standard deviation values for different existing methods with *TAG-RMSProp* for all four datasets. All metrics are averaged across 5 runs.

| Methods | Split-CIFAR100 | | Split-miniImageNet | |
|---|---|---|---|---|
| | Accuracy (%) | BWT (%) | Accuracy (%) | BWT (%) |
| *Naive SGD* | 51.36 ($\pm$3.21) | $-17.1$ ($\pm$2.64) | 48.19 ($\pm$0.79) | $-13.83$ ($\pm$1.97) |
| *Naive RMSProp* | 48.91 ($\pm$2.88) | $-18.37$ ($\pm$2.71) | 45.06 ($\pm$0.6) | $-19.32$ ($\pm$1.39) |
| *EWC* | 49.06 ($\pm$3.44) | $-17.76$ ($\pm$3.35) | 47.87 ($\pm$2.08) | $-13.79$ ($\pm$2.26) |
| *A-GEM* | 54.25 ($\pm$2.0) | $-14.73$ ($\pm$2.48) | 50.32 ($\pm$1.29) | $-10.69$ ($\pm$1.57) |
| *ER* | 59.14 ($\pm$1.77) | $-11.22$ ($\pm$2.19) | 54.67 ($\pm$0.71) | $-9.39$ ($\pm$0.64) |
| *Stable SGD* | 57.04 ($\pm$1.07) | $-7.59$ ($\pm$0.36) | 51.81 ($\pm$1.66) | $-8.18$ ($\pm$1.18) |
| **TAG-RMSProp (Ours)** | **62.79** ($\pm$0.29) | $-9.27$ ($\pm$1.16) | **57.2** ($\pm$1.37) | $-5.52$ ($\pm$1.71) |
| Methods | Split-CUB | | 5-dataset | |
| | Accuracy (%) | BWT (%) | Accuracy (%) | BWT (%) |
| *Naive SGD* | 54.88 ($\pm$1.83) | $-11.09$ ($\pm$1.43) | 46.48 ($\pm$3.62) | $-38.06$ ($\pm$3.69) |
| *Naive RMSProp* | 49.4 ($\pm$1.77) | $-22.36$ ($\pm$0.95) | 45.49 ($\pm$1.89) | $-40.09$ ($\pm$2.6) |
| *EWC* | 55.66 ($\pm$0.97) | $-10.7$ ($\pm$0.39) | 48.58 ($\pm$1.47) | $-30.98$ ($\pm$3.34) |
| *A-GEM* | 56.91 ($\pm$1.37) | $-8.69$ ($\pm$0.93) | 55.9 ($\pm$2.58) | $-26.71$ ($\pm$3.6) |
| *ER* | 59.25 ($\pm$0.82) | $-6.93$ ($\pm$0.92) | 61.58 ($\pm$2.65) | $-22.72$ ($\pm$3.08) |
| *Stable SGD* | 53.76 ($\pm$2.14) | $-8.39$ ($\pm$1.26) | 46.51 ($\pm$2.75) | $-36.79$ ($\pm$2.19) |
| **TAG-RMSProp (Ours)** | **61.58** ($\pm$1.24) | $-9.99$ ($\pm$1.62) | **62.59** ($\pm$1.82) | $-23.49$ ($\pm$1.73) |

### A.3.2 COMPARING *TAG-RMSProp* WITH OTHER BASELINES

Fig. 4 provides a detailed view of the test accuracies of individual baseline as the model encounters the new tasks throughout the LLL stream in **Split-CIFAR100**, **Split-miniImageNet** and **Split-CUB**. At the starting task $t = 1$, *TAG-RMSProp* beats other baselines because of lower initial learning rates (see Appendix A.2.1) and reflects the performance gain by the RMSProp over SGD optimizer. In all three datasets, the performance of *TAG-RMSProp* is very similar to *ER* specially from task $t = 5$ to task $t = 15$, but ultimately improves as observed at $t = 20$. These results show a decent gain in the final test **Accuracy** by *TAG-RMSProp* as compared to other baselines.

Although *TAG-RMSProp* is outperformed by *ER* from task $t = 2$ to $t = 4$, it results in the highest final accuracy as compared to other methods on **5-dataset** . *TAG-RMSProp* also outperforms other baselines including *Stable SGD* and *A-GEM* by a large margin.

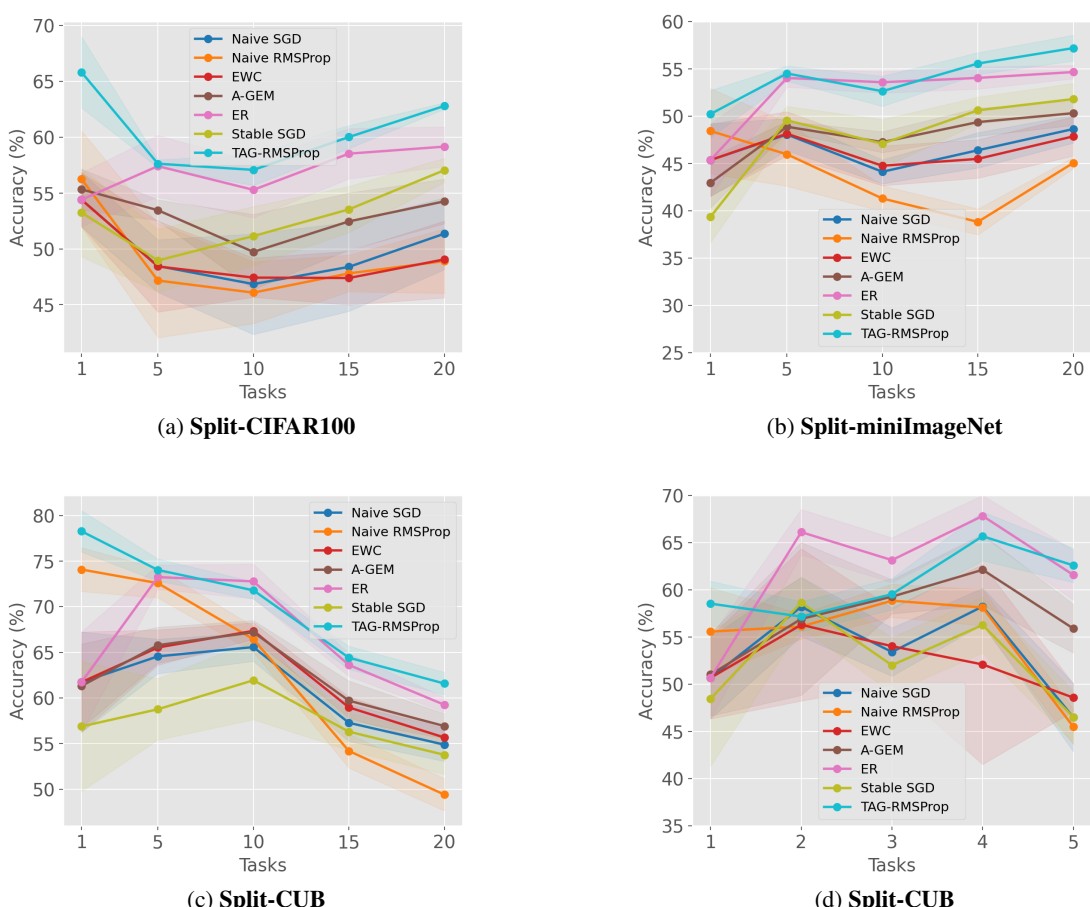

(a) **Split-CIFAR100**

(b) **Split-miniImageNet**

(c) **Split-CUB**

(d) **Split-CUB**

Figure 4: Evolution of average test **Accuracy (%)** $A_t$ for different existing methods and *TAG-RMSProp* throughout the stream on all datasets. All results are averaged across 5 runs and the shaded area represent standard deviation. Performing similar as *ER* for major part of the stream, *TAG-RMSProp* always results in the highest final accuracy as compared to other methods with a low standard deviation.

### A.3.3 EVOLUTION OF $\alpha(t, \tau)$ AND TEST ACCURACY $a_{t,\tau}$

Next, we continue the analysis done in Section **??** for **Split-CIFAR100** (in Fig. 5), **Split-miniImageNet** (in Fig. 6), **Split-CUB** (in Fig. 7) for the first 9 tasks and **5-dataset** (in Fig. 8) for first 3 tasks. In **Split-CIFAR100** and **Split-miniImageNet**, the model with *Naive RMSProp* tends to forget the task $t$ by significant amount as soon as it receives the new tasks. On the other hand, *TAG-RMSProp* prevents catastrophic forgetting and hence results in a higher accuracy throughout the stream. We can observe that for **Split-CIFAR100** and **Split-miniImageNet**, $\alpha(\tau + 1, \tau)$ (where $\tau \in [1, 9]$) generally have a higher rank in the set

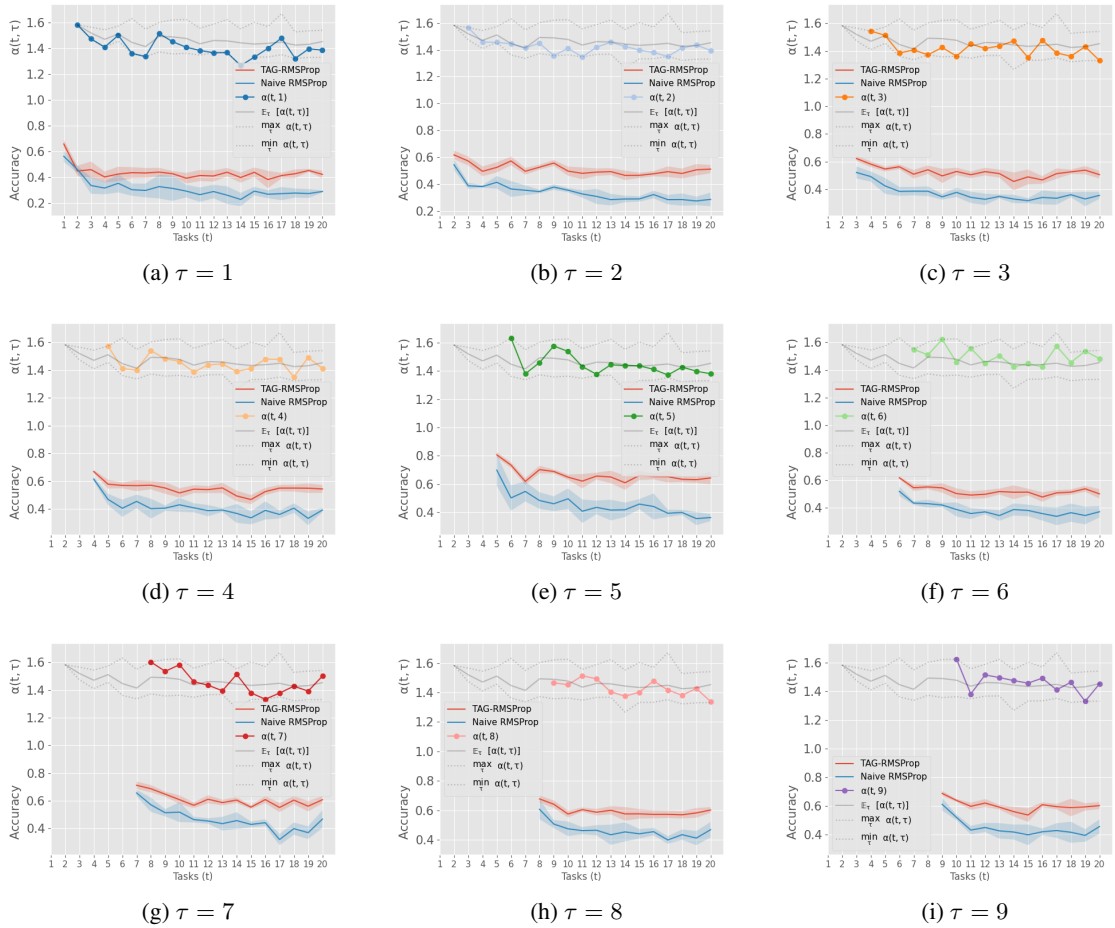

Figure 5: Evolution of $\alpha(t, \tau)$ and test accuracy $a_{t,\tau}$ where $\tau \in [1, 9]$ along the stream of 20 tasks in the **Split-CIFAR100** dataset. The grey-coloured lines are $\max_{\tau'} \alpha_n(t, \tau')$ (top, dashed line), $\mathbb{E}_{\tau'}[\alpha(t, \tau')]$ (middle, solid line) and $\min_{\tau'} \alpha(t, \tau')$ (bottom, dashed line) that indicate the range of $\alpha(t, \tau')$.

$\{\alpha(t, \tau'); \tau' \in [1, t]\}$. This is because *TAG-RMSProp* also recognizes an immediate change in the directions when the model receives a new task (from $M_N^{(t-1)}$ to $M_n^{(t)}$). A similar observation is made in case of **Split-CUB** but the visible gain in the accuracy by *TAG-RMSProp* does not occur instantly. Apart from that, we observe that the lower and higher rank of $\alpha(t, \tau)$ results in backward transfer and prevents catastrophic forgetting respectively in the stream. Overall, in all datasets, we arrive at the same conclusion obtained in Section **??**.

### A.3.4 MULTIPLE-PASS PER TASK

In this section, we report the performance of *TAG-RMSProp* and all other baselines discussed in Section 4.2 for 5 epochs per task in Table 5. Hyper-parameters for this experiment are kept the same as the single-pass per task setting. *TAG-RMSProp* results in high average Accuracy in all the datasets. We also observe less amount

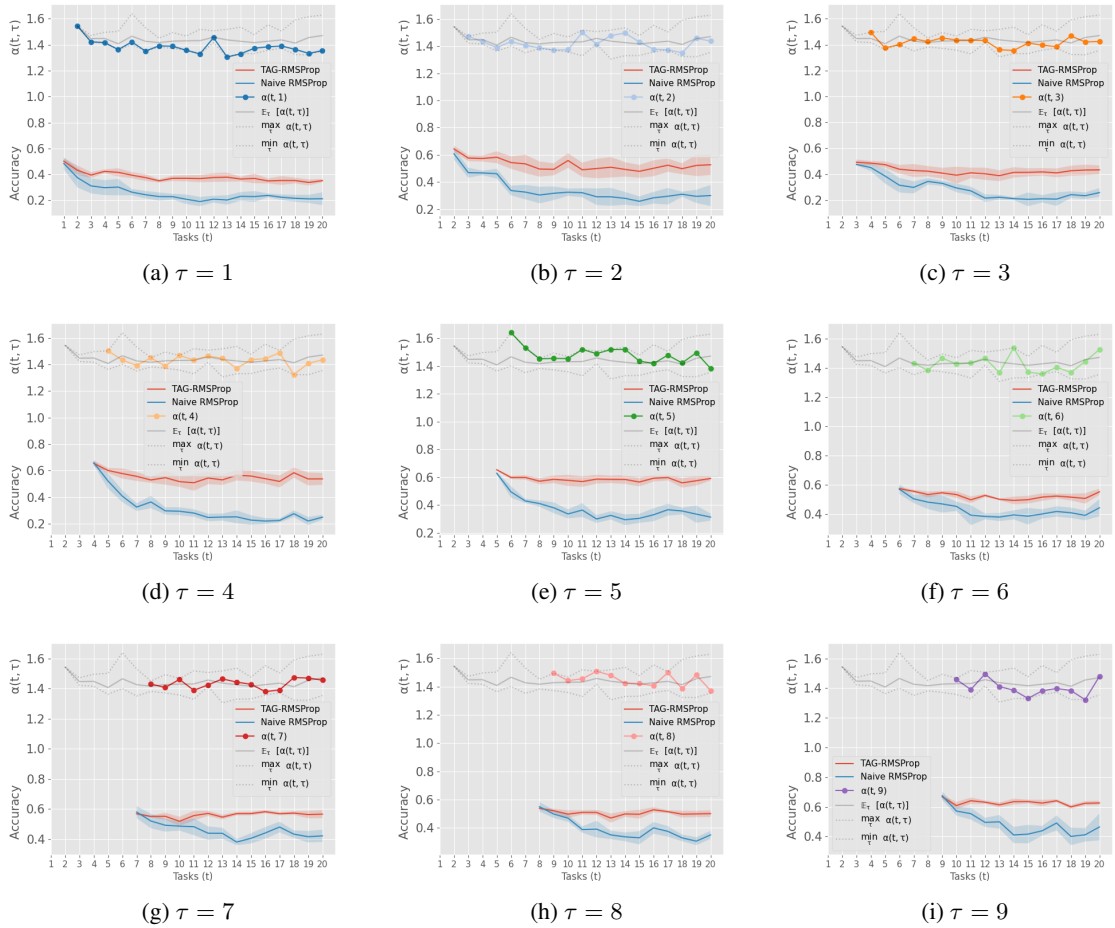

Figure 6: Evolution of $\alpha(t, \tau)$ and test accuracy $a_{t,\tau}$ where $\tau \in [1, 9]$ along the stream of 20 tasks in the **Split-miniImageNet** dataset. The grey-coloured lines are $\max_{\tau'} \alpha_n(t, \tau')$ (top, dashed line), $\mathbb{E}_{\tau'}[\alpha(t, \tau')]$ (middle, solid line) and $\min_{\tau'} \alpha(t, \tau')$ (bottom, dashed line) that indicate the range of $\alpha(t, \tau')$.

of **Forgetting** in *TAG-RMSProp* as compared to other baselines. In terms of **Learning Accuracy**, *TAG-RMSProp* is outperformed by the other baselines in **Split-CIFAR100**, **Split-miniImageNet** and **5-dataset** but performs better in **Split-CUB**.

### A.3.5 BIGGER MEMORY SIZE IN REPLAY-BASED METHODS

We also compare the performance of *A-GEM* and *ER* with a larger number of samples per class ($\mathbb{M}$) in the episodic memory for all four datasets in Table 6. With $\mathbb{M} = 10$, total episodic memory size for **Split-CIFAR100**, **Split-miniImageNet**, **Split-CUB** and **5-dataset** becomes 1000, 1000, 2000 and 500 respectively. We observe *ER* results in a significant gain in the performance as the episodic memory size increases. But *TAG-RMSProp* is able to outperform *A-GEM* in **Split-CIFAR100**, **Split-miniImageNet** and **Split-CUB** with a large margin even when $\mathbb{M}$ is set to 10.

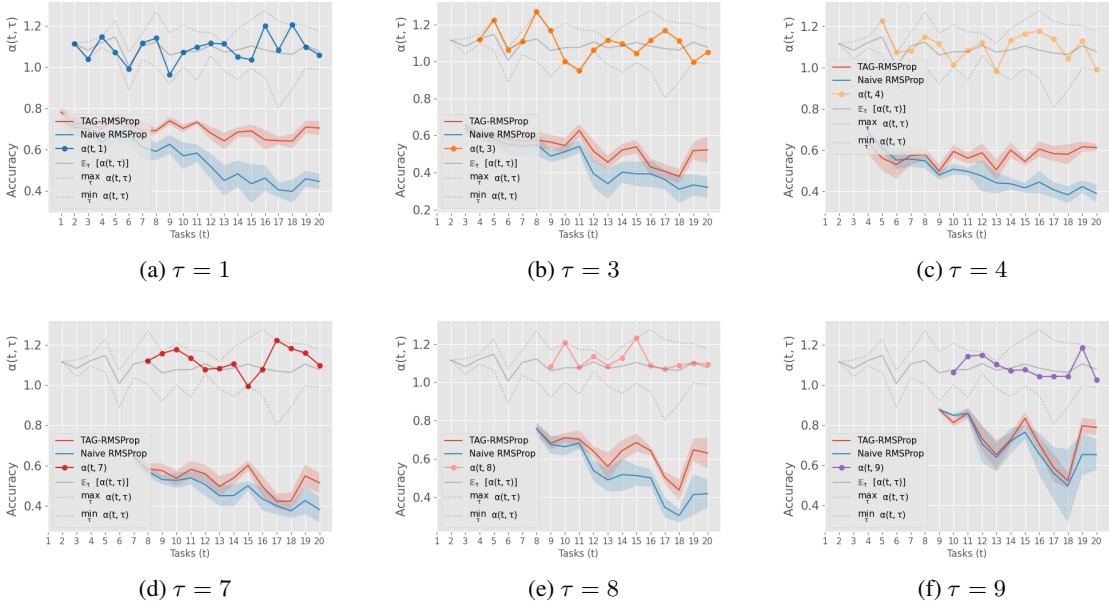

Figure 7: Evolution of $\alpha(t, \tau)$ and test accuracy $a_{t,\tau}$ where $\tau \in \{1, 3, 4, 7, 8, 9\}$ along the stream of 20 tasks in the **Split-CUB** dataset. The grey-coloured lines are $\max_{\tau'} \alpha_n(t, \tau')$ (top, dashed line), $\mathbb{E}_{\tau'}[\alpha(t, \tau')]$ (middle, solid line) and $\min_{\tau'} \alpha(t, \tau')$ (bottom, dashed line) that indicate the range of $\alpha(t, \tau')$.

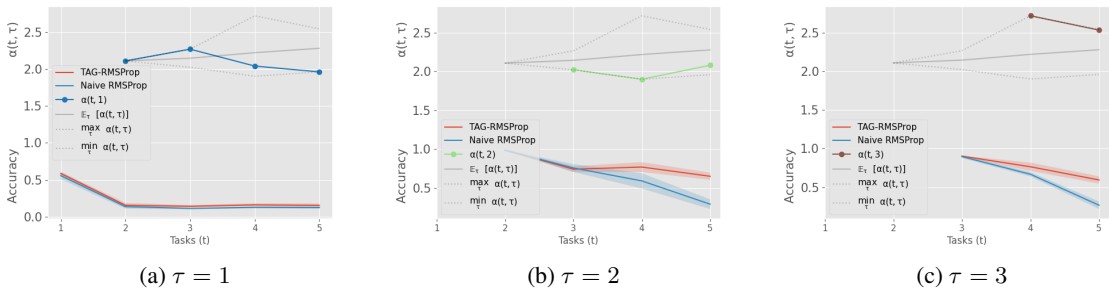

Figure 8: Evolution of $\alpha(t, \tau)$ and test accuracy $a_{t,\tau}$ where $\tau \in \{1, 2, 3\}$ along the stream of 5 tasks in the **5-dataset** dataset. The grey-coloured lines are $\max_{\tau'} \alpha_n(t, \tau')$ (top, dashed line), $\mathbb{E}_{\tau'}[\alpha(t, \tau')]$ (middle, solid line) and $\min_{\tau'} \alpha(t, \tau')$ (bottom, dashed line) that indicate the range of $\alpha(t, \tau')$.

Table 5: Comparing performance in terms of Final average test **Accuracy (%)** (higher is better), **Forgetting** (lower is better) and Learning Accuracy (**LA (%)**) (higher is better) with the standard deviation values for different existing methods with *TAG-RMSProp* running for **5 epochs per task** for all four datasets (see Section 4). All metrics are averaged across 5 runs. *MTL* assumes that the dataset from all tasks are always available during training, hence it is a different setting and its accuracy acts as an upper bound.

| Methods | Split-CIFAR100 | | | Split-miniImageNet | | |
|---|---|---|---|---|---|---|
| | Accuracy (%) | Forgetting | LA (%) | Accuracy (%) | Forgetting | LA (%) |
| *Naive SGD* | 52.26 (±0.65) | 0.28 (±0.01) | 78.45 (±0.41) | 46.9 (±1.18) | 0.25 (±0.02) | 70.82 (±0.6) |
| *Naive RMSProp* | 46.12 (±2.33) | 0.32 (±0.02) | 76.29 (±0.53) | 41.07 (±0.66) | 0.32 (±0.01) | 71.43 (±0.42) |
| *EWC* | 51.7 (±1.71) | 0.27 (±0.02) | 77.72 (±0.84) | 48.17 (±0.81) | 0.25 (±0.01) | 71.87 (±0.26) |
| *A-GEM* | 54.24 (±1.14) | 0.25 (±0.01) | 78.38 (±0.39) | 49.08 (±0.52) | 0.23 (±0.01) | 70.49 (±0.4) |
| *ER* | 60.03 (±0.96) | 0.19 (±0.01) | 78.15 (±0.7) | 54.01 (±0.56) | 0.19 (±0.01) | 71.77 (±0.58) |
| *Stable SGD* | 58.92 (±0.73) | 0.19 (±0.01) | 76.91 (±0.72) | 51.23 (±0.88) | 0.22 (±0.01) | 71.77 (±0.56) |
| **TAG-RMSProp (Ours)** | 60.64 (±1.38) | 0.17 (±0.01) | 77.12 (±0.76) | **58.0** (±1.11) | 0.11 (±0.02) | 68.14 (±0.38) |
| *MTL** | 67.7 (±0.58) | - | - | 66.14 (±1.0) | - | - |
| Methods | Split-CUB | | | 5-dataset | | |
| | Accuracy (%) | Forgetting | LA (%) | Accuracy (%) | Forgetting | LA (%) |
| *Naive SGD* | 59.87 (±1.48) | 0.21 (±0.02) | 79.77 (±0.44) | 49.95 (±2.42) | 0.51 (±0.04) | 90.86 (±0.63) |
| *Naive RMSProp* | 35.87 (±1.14) | 0.46 (±0.01) | 79.59 (±0.3) | 50.47 (±0.99) | 0.51 (±0.01) | 90.89 (±0.44) |
| *EWC* | 59.73 (±2.4) | 0.21 (±0.02) | 79.8 (±0.58) | 52.51 (±7.34) | 0.43 (±0.09) | 86.8 (±2.52) |
| *A-GEM* | 62.65 (±1.61) | 0.17 (±0.02) | 79.1 (±0.4) | 62.48 (±3.16) | 0.35 (±0.04) | 90.53 (±0.73) |
| *ER* | 66.06 (±1.28) | 0.14 (±0.02) | 78.79 (±0.55) | 62.84 (±1.58) | 0.35 (±0.02) | 90.52 (±0.69) |
| *Stable SGD* | 58.75 (±0.96) | 0.19 (±0.01) | 76.6 (±0.64) | 51.95 (±3.83) | 0.48 (±0.05) | 90.41 (±0.29) |
| **TAG-RMSProp (Ours)** | **68.0** (±1.01) | 0.13 (±0.01) | 80.15 (±0.22) | 61.13 (±3.05) | 0.36 (±0.04) | 89.9 (±0.33) |
| *MTL** | 71.65 (±0.76) | - | - | 70.0 (±4.44) | - | - |

Table 6: Comparing performance in terms of Final average test **Accuracy (%)** (higher is better), **Forgetting** (lower is better) and Learning Accuracy (**LA (%)**) (higher is better) with the standard deviation values for *A-GEM* and *ER* for different number of samples per class ($\mathbb{M}$) in the episodic memory with *TAG-RMSProp* for all four datasets (see Section 4). All metrics are averaged across 5 runs. Overall, *ER* with bigger memory outperforms all other methods in terms of Accuracy.

| Methods | Split-CIFAR100 | | | Split-miniImageNet | | |
|---|---|---|---|---|---|---|
| | Accuracy (%) | Forgetting | LA (%) | Accuracy (%) | Forgetting | LA (%) |
| *A-GEM* ($\mathbb{M} = 1$) | 54.25 (±2.0) | 0.16 (±0.03) | 68.98 (±1.19) | 50.32 (±1.29) | 0.11 (±0.02) | 61.02 (±0.64) |
| *A-GEM* ($\mathbb{M} = 5$) | 55.74 (±1.14) | 0.14 (±0.01) | 68.97 (±0.56) | 49.52 (±2.02) | 0.12 (±0.02) | 60.49 (±0.78) |
| *A-GEM* ($\mathbb{M} = 10$) | 56.68 (±1.92) | 0.13 (±0.02) | 68.72 (±0.96) | 49.77 (±2.41) | 0.12 (±0.02) | 60.6 (±0.66) |
| *ER* ($\mathbb{M} = 1$) | 59.14 (±1.77) | 0.12 (±0.02) | 70.36 (±1.23) | 52.76 (±1.53) | 0.1 (±0.01) | 61.7 (±0.74) |
| *ER* ($\mathbb{M} = 5$) | 65.74 (±1.47) | 0.07 (±0.01) | 70.91 (±1.13) | 58.49 (±1.21) | 0.05 (±0.01) | 62.24 (±0.85) |
| *ER* ($\mathbb{M} = 10$) | **68.94** (±0.93) | 0.05 (±0.01) | 71.26 (±1.01) | **60.06** (±0.63) | 0.04 (±0.01) | 62.21 (±1.24) |
| **TAG-RMSProp (Ours)** | 62.79 (±0.29) | 0.1 (±0.01) | 72.06 (±1.01) | 57.2 (±1.37) | 0.06 (±0.02) | 62.73 (±0.61) |
| Methods | Split-CUB | | | 5-dataset | | |
| | Accuracy (%) | Forgetting | LA (%) | Accuracy (%) | Forgetting | LA (%) |
| *A-GEM* ($\mathbb{M} = 1$) | 56.91 (±1.37) | 0.1 (±0.01) | 65.6 (±0.73) | 55.9 (±2.58) | 0.34 (±0.04) | 82.61 (±2.13) |
| *A-GEM* ($\mathbb{M} = 5$) | 56.4 (±1.5) | 0.1 (±0.01) | 65.63 (±0.64) | 61.39 (±1.0) | 0.28 (±0.01) | 83.48 (±1.05) |
| *A-GEM* ($\mathbb{M} = 10$) | 56.71 (±1.6) | 0.1 (±0.01) | 65.73 (±0.9) | 62.43 (±1.38) | 0.26 (±0.03) | 83.38 (±1.79) |
| *ER* ($\mathbb{M} = 1$) | 59.25 (±0.82) | 0.1 (±0.01) | 66.17 (±0.42) | 61.58 (±2.65) | 0.28 (±0.04) | 84.31 (±1.08) |
| *ER* ($\mathbb{M} = 5$) | 68.89 (±0.31) | 0.04 (±0.0) | 66.76 (±0.73) | 71.56 (±1.54) | 0.16 (±0.02) | 84.34 (±1.46) |
| *ER* ($\mathbb{M} = 10$) | **70.73** (±0.23) | 0.03 (±0.01) | 66.83 (±0.86) | **75.44** (±1.07) | 0.12 (±0.02) | 84.62 (±0.89) |
| **TAG-RMSProp (Ours)** | 61.58 (±1.24) | 0.11 (±0.01) | 71.56 (±0.74) | 62.59 (±1.82) | 0.3 (±0.02) | 86.08 (±0.55) |

