# OpenReview forum: "TAG: Task-based Accumulated Gradients for Lifelong learning"
_ICLR.cc/2022/Conference — ICLR 2022 Submitted_

### Official Review · Reviewer_m25m · 2021-10-22

**Correctness:** 4
**Technical Novelty And Significance:** 2
**Empirical Novelty And Significance:** 3
**Recommendation:** 5
**Confidence:** 4

**Main Review:**

Strength:

*  The paper is clearly written and easy to follow
*  Experiments on different settings are considered.


Weakness:
*  The technical novelty of the proposed method is limited. In fact, TAG is similar to LA-MAML [1] (Gupta et al.,2020). Both the motivations and derived formula (Equation 2) of TAG are very similar to [1]. From this aspect, TAG can be seen as a memory-free version of LA-MAML.

* misinterpretation of related works.  The author claims that “Another related work (Gupta et al.,2020) also employs an adaptive learning rate while requiring a small episodic memory. But it is based on a
meta-learning setting and hence beyond the scope of our paper”. This is an incorrect statement, LA-MAML  (Gupta et al.,2020) is to solve the continual learning problems with the tools from  meta learning. They are not in meta-learning setting, but instead in the continual learning setting.

*  The proposed method seems to only work on the task-aware setting. More recent works focus on the task-free continual learning setting, where the task identity and boundary are unknown during the learning process.  It would strengthen the proposed method if  TAG can be applied in such scenarios.

*  The experiments on more other architectures, such as MLP may demonstrate the effectiveness of the proposed method.







**Summary Of The Paper:**

This paper proposes a  task-aware adaptive learning rate method, TAG, for continual learning. The optimizer TAG is to promote the learning rate if they are similar to previous tasks, while decreasing the learning rates if they are dissimilar to previous tasks without storing previous examples. The authors combined the proposed method with existing optimizers, such as Adam, SGD, etc to demonstrate the effectiveness of the proposed method. Experimental results on several datasets show the improvements over naive optimizers.


**Summary Of The Review:**

The method novelty is limited, and experiment needs to be improved.

---

> ### Author Response · Authors · 2021-11-19
> **Response to Reviewer m25m**
>
> We’d like to thank the reviewer for the comments on our paper. We provide our point-by-point responses below:
>
> **Reviewer’s Comment: “TAG is similar to LA-MAML [1] (Gupta et al.,2020). Both the motivations and derived formula (Equation 2) of TAG are very similar to [1]. From this aspect, TAG can be seen as a memory-free version of LA-MAML”**
>
> **Response:** We agree that TAG is similar to LA-MAML with the motivation that the learning rate can help reflect the similarities between the old and new tasks. But, we would also like to point out that the key idea for TAG is to additively accumulate the gradients that the network took while learning a specific task. Since, the *similarity* between adaptation trajectories was shown as a good indicator for generalization (Guiroy et al., 2019 [2]), we used the task-based first moment (a less noisy estimate for the overall gradient direction) to capture the task-specific gradient directions without storing the past examples. Hence, TAG explicitly computes the similarities between the tasks and further uses them to regulate the effect of task-based second moments in the learning rate.
>
> **Reviewer’s Comment: “LA-MAML (Gupta et al.,2020) is to solve the continual learning problems with the tools from meta learning. They are not in meta-learning setting, but instead in the continual learning setting.”**
>
> **Response:** We would like to thank the reviewer for pointing it out and assure that we will correct the interpretation in the final manuscript.
>
> When we compare La-MAML with TAG on Split-CIFAR100, the final Accuracy (%) and BWT (%) looks like the following:
>
> **Methods: | Accuracy | BWT**
>
> La-MAML (Reported in their paper) | 61.18 (± 1.4) | -9.00 (± 0.2)
>
> TAG-RMSProp | 62.79 (±0.29) | −9.27 (±1.16)
>
> We will also add the detailed experiment and results in the final manuscript.
>
> **Reviewer’s Comment: “The proposed method seems to only work on the task-aware setting”**
>
> **Response:** We appreciate the suggestion made by the reviewer on TAG to be applied on task-free continual learning settings (or class-incremental settings). This would be an interesting extension of our work. One of the possible ways to do it would be by clustering the tasks together (based on the computed similarity) in settings with soft margins between the tasks.
>
> We would also like to highlight that a task-aware setting (or task-incremental learning) is one of the three standard scenarios in continual learning and numerous state-of-the-art works in the literature just focus on task-incremental learning.
>
> **Reviewer’s Comment: “The experiments on more other architectures, such as MLP may demonstrate the effectiveness of the proposed method. ”**
>
> **Response:** We appreciate the reviewer’s suggestion of using architectures for the experiments. Therefore, we experiment on Rotated-MNIST using MLP with two hidden layers (each with 400 units). We construct this dataset for 10 tasks (with 30-degree rotation in each). The resulting final Accuracy (%) and LA (%) looks like the following:
>
> **Methods: | Accuracy**
>
> Naive SGD |	77.9 (± 0.9)
>
> ER (with 1 example per class) |	80.1 (± 1.2)
>
> ER (with 10 example per class)|	86.9 (± 1.0)
>
> TAG-RMSProp | 89.5 (± 0.7)
>
> We will add the details of this experiment in the final manuscript.
>
> [2] Simon Guiroy, Vikas Verma, and Christopher Pal. Towards understanding generalization in gradient-based meta-learning. arXiv preprint arXiv:1907.07287, 2019.

---

> > ### Comment · Reviewer_m25m · 2021-11-29
> > **Response to author feedback**
> >
> > Thank you for the author's feedback and additional experiments. I appreciate the author's efforts. I think the main concerns are still the limited technical novelty, so I have to keep my score.

---

### Official Review · Reviewer_94tf · 2021-11-01

**Correctness:** 2
**Technical Novelty And Significance:** 3
**Empirical Novelty And Significance:** 2
**Recommendation:** 5
**Confidence:** 4

**Main Review:**

Strenghts of this paper are:
- The method provides a simple yet effective way of dealing with catastrophic forgetting, which is to some extent original since it proposes to use the relatedness among tasks and use this information to control gradient updates while learning new tasks.
- The method is technically sound, and most of the experimental evaluation is aligned with typical evaluations in the area, including datasets used and reported metrics.
- The paper is well-written, well-structured and easy to follow.

Weaknesses of this paper are:
- My main concern is regarding memory/storage size requirements of the proposed method (during training). Although the proposed method avoids the need to keep examples of previous tasks, as compared to replay-based methods, and of expanding the network, compared to network expansion methods, it certainly adds memory/storage requirements of the gradients. Based on this, I would expect to see comparisons of the proposed method vs. replay-based methods and network expansion methods, in terms of memory usage.  I think that these concerns need to be fully addressed in this paper, to really demonstrate the advantages of the proposed approach. Furthermore, I would expect to see these comparisons for a reasonably large number of tasks, since memory/storage requirements increase along with the number of tasks.
- My second biggest concern is where are the actual gains of the proposed approach. As I can infer from Table 1, most of the gain in final accuracy is due to a gain in LA. According to the definition of LA, this is actually the accuracy of each new task. Therefore, I do not agree with the claim in page 8: "The higher LA with similar Forgetting as compared to other baselines shows that while TAG exploits the adaptive nature of existing optimizers, it also ensures minimal forgetting of the gained knowledge... Hence, even if a similar (or lower) Forgetting occurs in TAG, the higher test Accuracy (with high LA) shows that TAG is capable of retaining the gained knowledge from each task.", since from Table 1 forgetting levels are actually very similar to those experienced by other methods and therefore a higher accuracy (which I agree is due to LA) comes mainly from learning new tasks better rather than encouraging "minimal forgetting" or "retaining the gained knowledge" as it is claimed. I would strongly suggest to clarify this point.
- In the experiments, I do not see the reason for not comparing with network expansion methods. The fact that the proposed method is not making any change to the size of the model, as mentioned in page 8, does not create any limitation for comparing with these kinds of methods in the experimental setting used in the paper.
- Finally, I see a lot of similarities of the proposed method with existing methods to control gradient updates such as OGD [1] and OWM [2]. Therefore, I would expect comparisons to these methods.

[1] Farajtabar, M., Azizan, N., Mott, A., & Li, A. (2020, June). Orthogonal gradient descent for continual learning. In International Conference on Artificial Intelligence and Statistics (pp. 3762-3773). PMLR.

[2] Zeng, G., Chen, Y., Cui, B., & Yu, S. (2019). Continual learning of context-dependent processing in neural networks. Nature Machine Intelligence, 1(8), 364-372.

**Summary Of The Paper:**

This paper proposes TAG, a method for continual learning in the task-incremental setting. This method relies on storing and using task gradients while learning a set of supervisd tasks sequentially. The influence task-based accumulated gradients is regulated through a learning rate that is adaptive according to the relatedness of the current task with the previously observed ones. The authors report results on benchmark datasets for continual learning of up to 20 disjoint tasks. Comparisons against naive non-continual optimizers such as SGD, Adam and RMSProp are reported, along with results in the continual learning setting with some state-of-the-art methods such as EWC, A-GEM and ER. The authors report performance in terms of overall accuracy, forward transfer and learning accuracy (LA).

**Summary Of The Review:**

Although the paper proposes a simple method for controlling catastrophic forgetting in the continual learning setting, there are major drawbacks flaws in the experimental results and the evaluation of the proposed approach. Given the nature of TAG, I think is is fundamental to measure and report memory/storage requirements in comparison to other methods that also require additional memory/storage such as replay-based methods and network expansion methods. Furthermore, claims around avoidance of catastrophic forgetting in the main results reported in Table 1 and explained in page 8 are not well-supported since it is clear that the method is strong at learning new tasks, while not necessarily much better than counter parts regarding catrastrophic forgetting.

---

> ### Author Response · Authors · 2021-11-19
> **Response to Reviewer 94tf**
>
> We would like to thank the reviewer for the comments on our paper. We provide our point-by-point responses below:
>
> **Reviewer’s Comment: “Although the proposed method avoids the need to keep examples of previous tasks, as compared to replay-based methods, and of expanding the network, compared to network expansion methods, it certainly adds memory/storage requirements of the gradients. Based on this, I would expect to see comparisons of the proposed method vs. replay-based methods and network expansion methods, in terms of memory usage”**
>
> **Response:** Upon the reviewer’s suggestions, we will add the comparisons between TAG with replay-based methods and network expansion methods in terms of memory usage. We would like to highlight that our proposed method requires extra memory only for the optimizer and not for the model itself. On the other hand, network expansion methods like Progressive Neural Networks [3] require quadratic growth in the number of parameters as the number of tasks increases. We also would like to highlight that Orthogonal Gradient Descent (OGD) requires storing 200 gradients per task on MNIST datasets. Since TAG additively accumulates the gradients and hence requires memory equal to two copies of the model as the knowledge base, OGD is 100 times costlier than our approach.
>
>
> **Reviewer’s Comment: “a higher accuracy (which I agree is due to LA) comes mainly from learning new tasks better rather than encouraging "minimal forgetting" or "retaining the gained knowledge" as it is claimed”**
>
> **Response:** We respectfully disagree. Apart from attaining high LA, we would like to highlight that TAG achieves a higher accuracy by encouraging minimal forgetting and retaining the gained knowledge. For example, in terms of LA, TAG performs similar to (or even outperformed by) methods like Naive RMSProp (Split-CUB) and ER (Split-miniImageNet). But, TAG still ensures minimal forgetting of the gained knowledge and achieves higher accuracy than these methods. The analysis presented in Figure 2 (Section 4.1) where the higher rank of $\alpha(t,\tau)$ results in preventing forgetting also validates the same.
>
> **Reviewer’s Comment: “I see a lot of similarities of the proposed method with existing methods to control gradient updates such as OGD [1] and OWM [2]. Therefore, I would expect comparisons to these methods”**
>
> **Response:** Although there are similarities of the proposed method in terms of controlling the gradient updates, OGD requires storing a large number of gradients per task for projecting to the space of previous model gradients. On the other hand, memory equal to two copies of the model in TAG enabled us to train a Reduced Resnet-18 on complex datasets. Due to greater memory requirements, OGD faced memory errors in our setting. Therefore, we experimented with Split-CIFAR100 and considered N=50 for OGD. The final Accuracy (%) and LA (%) looks like the following:
>
> **Methods: | Accuracy (LA)**
>
> OGD (with 1 epoch) |	29.87 (49.10)
>
> OGD (with 5 epochs) |	41.8 (64.95)
>
> ER |	59.14 (70.36)
>
> TAG-RMSProp | 62.79 (72.06)
>
> It is also important to note that OGD requires multiple epochs to learn and hence is not suitable for the online setup that we focus on.
>
> We also thank the reviewer for pointing us to OWM. We will add a comparison to OGD and OWM in the final version of the paper. We would like to highlight that both OGD and OWM use simple SGD and SGD with momentum optimizers respectively. Our contribution is an adaptive learning rate based method for lifelong learning which in theory is complementary to both these approaches.
>
>
> [3] Rusu, Andrei A., et al. "Progressive neural networks." arXiv preprint arXiv:1606.04671 (2016).

---

> > ### Comment · Reviewer_94tf · 2021-11-29
> > **Response to authors**
> >
> > Thanks to the authors for answering most of my concerns, and for the additional explanations. However, I am still missing actual clear results for the evaluated datasets in terms of memory usage, especially vs. example replay methods. Based on this and on others reviews, I can only increase my score to 'below acceptance threshold'.

---

### Official Review · Reviewer_ZR5n · 2021-11-02

**Correctness:** 4
**Technical Novelty And Significance:** 2
**Empirical Novelty And Significance:** 3
**Recommendation:** 8
**Confidence:** 4

**Main Review:**

Pros:
* General approach that can be adapted to multiple optimization algorithms and used in other meta-algorithms
* Strong performance in a single-pass scenario
* Can achieve better learning accuracy than naive methods

Cons:
* The effect of the proposed procedure on optimization algorithms is not studied enough. For example, it’s unclear how would a few similar tasks help to learn faster when there are many distinct tasks present. Additional insights, like visualizations of trajectories, will help to build an intuition and boost the method adoption in practice.
* Effect of task order is not studied/highlighted: the method seems to be very dependent on the task order and it is unclear how the order is chosen in experiments.

Minor comments/questions:
* what is the point of having \lambda(t,t) in eq. 3, i.e. why do you need to multiply the currently accumulated gradients by task similarity to itself?
* In appendix 3.3, Figure 5(a) there is a noticable difference between RMSProp and TAG-RMSProp on the first task. What is the reason for this? Shouldn’t they be identical/close?

**Summary Of The Paper:**

The authors present an approach for lifelong learning where each task is processed in a single pass. They propose to adapt the learning rate of the training algorithm depending on the current task's similarity to the previously observed tasks. The learning rate is decreased when there are many dissimilar tasks to avoid catastrophic forgetting. The paper presents an adapted RMSProp algorithm, but the procedure can be adjusted to Adagrad and Adam as well. The presented evaluation on computer vision datasets shows that the proposed modification helps to increase the final accuracy of those methods while keeping the forgetting low. When compared to baselines from literature under the conditions of the single pass setting, the proposed method achieves better final accuracy and keeps approximately the same forgetting rate.

**Summary Of The Review:**

The presented method is a general procedure that helps to improve performance of many methods in lifelong learning scenario. The presented experimental results are convicing. However, the paper would benefit from more research and intuition on why the approach works. I recommend to accept the paper.

---

> ### Author Response · Authors · 2021-11-19
> **Response to Reviewer ZR5n**
>
> We’d like to thank the reviewer for the positive evaluation of our paper. We also appreciate the comments and provide our point-by-point responses below:
>
> **Reviewer’s Comment: “how would a few similar tasks help to learn faster when there are many distinct tasks present. Additional insights, like visualizations of trajectories, will help to build an intuition and boost the method adoption in practice”**
>
> **Response:** The primary advantage of using an adaptive learning rate inspired by the existing ones is that it helps learn the new task faster. But TAG doesn’t completely prevent the new tasks updates from happening when the tasks are not similar. TAG essentially limits the drastic changes per-parameter to prevent catastrophic forgetting by putting weights ($\alpha(t, \tau)$) on task-based second moments $V_N^{(\tau)}$. Since these $\alpha(t, \tau)$ are defined using the hyperparameter $b$, their values lie within $[e^{-b}, e^{b}]$. Therefore, even if the tasks are negatively correlated (Correlation = -1 and $\alpha(t, \tau)=e^b$), a controlled update on parameters will happen depending on the value of $b$.
>
> We would also like to thank the reviewer for the suggestion to add visualizations of trajectories and assure that we will add it to build more intuition in the final version of the manuscript.
>
>
> **Reviewer’s Comment: “Effect of task order is not studied/highlighted”**
>
> **Response:** The task order in our experiments was chosen randomly as used by the existing baselines in their implementation. Although TAG's updates rely on the task-based first/second moments, the task-order has little to no effect on the overall performance since TAG can outperform other baselines on multiple complex datasets with a randomly chosen task order. Moreover, TAG’s new updates also rely on the correlation of the current task with the previous tasks irrespective of the order. To validate it empirically, we will perform the experiment for investigating the effect of task-order and add the results in the final version of the manuscript.
>
> **Reviewer’s Comment: “what is the point of having \lambda(t,t) in eq. 3,”**
>
> **Response:** The $\alpha(t, t)$ (directly computed as $e^{-b}$) in Eq. 3 is used for uniform (and unbiased) weight assignment on $V_N^{(\tau)}$.
>
> **Reviewer’s Comment: “In appendix 3.3, Figure 5(a) there is a noticable difference between RMSProp and TAG-RMSProp on the first task.”**
>
> **Response:** In appendix 3.3, Figure 5(a), since the hyperparameters setting was found by performing grid-search, the optimal learning rate resulting in the best final performance for RMSProp was different from TAG-RMSProp. Therefore, the task 1 accuracies in these two methods are different.

---

> > ### Comment · Reviewer_ZR5n · 2021-11-26
> > **Re: response**
> >
> > Thank you for the explanations.
> >
> > **Re: task order:** I am not suggesting that the task order will change the results of your comparison to the baselines, but rather that there can be more and less favorable task orderings for TAG and it would have been interesting to understand when to does better, e.g. when there are many similar tasks in a row or when the task are mixed between similar and dissimilar. It's not crucial for the paper in the current form, but can provide better insight in TAG's update behavior.
> >
> > **Recommendation:**
> > I think the authors addressed the feedback from all reviews and I still recommend to accept the submission.

---

### Official Review · Reviewer_qvJd · 2021-11-08

**Correctness:** 3
**Technical Novelty And Significance:** 3
**Empirical Novelty And Significance:** 2
**Recommendation:** 5
**Confidence:** 3

**Main Review:**

I think the idea is clear, and the experiments verify the effectiveness of the proposed method. However, I found the results of A-GEM in this paper are significantly worse than the original paper. For example, the accuracy of split-CIFAR100 is 54.25% in this submission, while in A-GEM the accuracy is ~62%. Could you provide more details?

In addition, though the submission addresses the catastrophic forgetting by proposing a new optimizer, the discussion of task correlations are very similar to GEM and its variants (e.g., A-GEM) and CLAW (CONTINUAL LEARNING WITH ADAPTIVE WEIGHTS). More analysis about why the proposed method can outperform other approaches would make the paper more convincing.

Missing reference: [1] is a very relevant work. I think it might be valuable to compare and discuss with it.

Btw, the authors change the margin of the ICLR format. Please kindly revise it in the next version.

I am willing to adjust my score if the authors can address my concerns.

[1] Guo, Yunhui, Mingrui Liu, Tianbao Yang, and Tajana Rosing. "Improved schemes for episodic memory-based lifelong learning." arXiv preprint arXiv:1909.11763 (2019).


---After rebuttal ---

The authors' response does partially address my concerns. After reading the authors' rebuttal and other reviewers' comments, I still think the contributions are not enough for publication. I will keep my score.

**Summary Of The Paper:**

In this paper, the authors propose a new optimization method for continual learning. The authors propose a task-aware optimizer to adapt the learning rate for each task. The proposed method is evaluated on several datasets to show its effectiveness.


**Summary Of The Review:**

The idea is clear and the results seem promising. However, the authors are supposed to provide more analysis to understand why the proposed method works.

---

> ### Author Response · Authors · 2021-11-19
> **Response to Reviewer qvJd**
>
> We would like to thank the reviewer for the comments on our paper. We provide our point-by-point responses below:
>
> **Reviewer’s Comment: “results of A-GEM in this paper are significantly worse than the original paper.”**
>
> **Response:** The results of A-GEM in our paper are different because authors of A-GEM present their results for different memory sizes (65 per task for CIFAR) and also different learning rate as compared to ours.
>
>
> **Reviewer’s Comment: “the discussion of task correlations are very similar to GEM and its variants (e.g., A-GEM) and CLAW.”**
>
> **Response:**  We thank the reviewer for this comment which gives us an opportunity for clarification. Since the loss surface changes with each upcoming new task, we aim to benefit from the relatedness of gradient directions of tasks by accumulating them instead of computing them individually using examples from episodic memory (A-GEM) or differentiating between the task-based weights and the globally shared weights (CLAW) and applying projected gradient descent.
>
> We regulate the role of task-based second moments on the overall update on each parameter. We assign an “unrelatedness* weight ($\alpha(t,\tau)$) over them to limit the new updates if the tasks ($t$ and $\tau$) were unrelated. We compute the *unrelatedness* using the gradient directions from each task. We increase the *unrelatedness* value corresponding to the tasks and, as a result, decrease the magnitude of such updates. We limit (or scale down) the updates so that the parameters do not move far away from a previous optimum. Therefore, the model still learns a new task fast (as seen in high LA with only one epoch) and achieves high accuracy compared to existing baselines (including A-GEM).
>
>
> **Reviewer’s Comment: “Missing reference: ‘Improved schemes for episodic memory-based lifelong learning’ is a very relevant work. I think it might be valuable to compare and discuss with it.”**
>
> **Response:** We thank the reviewer for this suggestion. We will cite this reference in the final version of the manuscript and add a comparison.
>
> **Reviewer’s Comment: “the authors change the margin of the ICLR format.”**
>
> **Response:** We would like to thank the reviewer for pointing it out. We will make it correct in the final version of the manuscript.

---

### Decision · Program_Chairs · 2022-01-20

**Decision:**

Reject

**Comment:**

The authors develop a memory-based method for continual learning that stores gradient information from past tasks. This memory is then used by a proposed task-aware optimizer that, based on the task relatedness, aims at preserving knowledge learned in previous tasks.

The initial reviews were reasonable but indicated that this paper was not yet ready to be published. In particular, the reviewers seemed to agree on the somewhat limited methodological novelty of the paper given prior work (such as LA-MAML and OGD in terms of method and GEM in terms of task similarity comparison).

In their response, the authors do seem to agree to a certain extent with some of the criticisms, but also point to clear differences with respect to previous work (and other distinguishing aspects such as a smaller memory footprint than OGD). The authors also carefully responded to reviewer comments and provided additional results when possible.

In the end, the main criticism from the reviewers remained (Reviewer 95tf also suggests that the authors should compare their method to others in terms of memory consumption (which the authors partly did) and compare to replay-based methods) and this paper was a borderline one. Three, out of the four, reviewers suggest that it is not ready to be published. One reviewer did give it a high score (8) but also understood the limitations raised by the other reviewers. As a result, my recommendation is that this paper falls below the acceptance threshold.

I am sorry that for this recommendation and I strongly suggest the authors consider the reviewer's suggestions in preparing the next version of this work. In particular, it seems like providing a full study of the memory usage of your approach vs. others as well as providing more insights about the "trajectory" (see the comment from ZR5n) might go a long way toward improving the paper.